# Polycentric Solutions for Groundwater Governance in Sub-Saharan Africa: Encouraging Institutional Artisanship in an Extended Ladder of Participation

Bryan Bruns

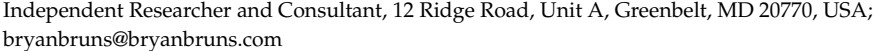

Independent Researcher and Consultant, 12 Ridge Road, Unit A, Greenbelt, MD 20770, USA; bryanbruns@bryanbruns.com

**Abstract:** The growth of groundwater irrigation poses opportunities and challenges, particularly in Africa where substantial potential exists for increased groundwater irrigation but has been constrained by limited access to energy, technology for pumps and drilling, markets, and other factors. Conventional groundwater governance concepts for state-led regulation or co-management are problematic for conditions where state capacity or political support for regulation to reconcile conflicting interests is limited. Experience in Africa and elsewhere does offer examples that may help recognize feasible patterns for collective action that can influence the equity, efficiency, and sustainability of groundwater development. An extended ladder of participation helps look beyond state-led water governance and co-management to a more diverse range of opportunities for supporting local autonomy and initiative to expand opportunities and solve problems in groundwater development. Collective action in groundwater governance can include well spacing; sharing of wells, pumps, and pipes; protecting domestic water sources; crop coordination; groundwater recharge; water imports; and aquifer management. Even where non-state organizations and collective action play primary roles in water governance, they may still be empowered by, receive advice from, or share information with government agencies and other actors. Polycentric groundwater governance can be supported by improving information, facilitating cooperation, endorsing standards, providing a legal framework for resolving conflicts and constituting governance agreements, and through polycentric social learning. Polycentric institutional artisanship by water users and their organizations can help find feasible solutions for improving groundwater governance.

**Keywords:** participatory groundwater governance; polycentric problem solving; institutional artisanship; farmer-led irrigation; community-based natural resource management; co-management; stakeholder engagement

## 1. Introduction

### 1.1. Potential for a Groundwater Irrigation Boom

There is increasing interest and enthusiasm regarding the potential of groundwater irrigation in Africa, including the prospects for solar-powered pumps [1–4]. Groundwater, alone or as part of a range of methods for agricultural water control, can help farmers grow more crops, coping with limited rainfall, drought, and climate change. Improved access to irrigation from groundwater can transform farmers' incomes and make their livelihoods more resilient. Research suggests many places in Africa have substantial amounts of potentially utilizable groundwater [5–10]. Experience in South Asia and elsewhere has shown how farmers' investment decisions can drive widespread application of groundwater pumping once conditions are favourable [4,11,12]. There is growing recognition of the extent to which farmers in Africa and elsewhere are taking a leading role in expanding irrigation, drawing on shallow and deep wells, wetlands, and other water sources often in ways not fully recognized by official statistics and policies [13,14].

*1.2. Threats from Inequity, Depletion, and Water Quality*

The potential for increased groundwater usage for irrigation also generates concern about possible problems. Those with advantages related to wealth, age, gender, political power, ethnic identity, or other social relationships might gain most benefits, with others excluded or marginalized, increasing social inequities. Entry points exist for making development more inclusive, such as through credit, market linkages, and focus on the concerns of women farmers, but may be difficult to implement and need further research and development [15].

While irrigation benefits farmers who pump, their groundwater abstraction can reduce availability of water in nearby areas and downstream in basins, affecting other water users and ecosystems. It is often hard for groundwater users to observe or understand how their use may affect others. Conditions underground are often not well understood, recharge processes can be complex, and impacts slow to appear. Variability in rainfall during the year, between years, and over longer time scales complicates understanding the impact of groundwater withdrawals. Institutions to cope with impacts of intensified water use (externalities) on others such as neighbours and downstream water users are often absent or ineffective. Water quality can also be a problem, especially where groundwater contains hazardous chemicals such as arsenic or fluorine that are not easily observable.

*1.3. Difficulties in Groundwater Governance*

Groundwater is usually extracted by many dispersed users, making usage difficult to monitor and control. Groundwater is a common pool resource, where one person's use subtracts from what is available for others, but it is difficult to restrict access [16]. Conventional approaches to top-down licensing and control over groundwater abstraction have often been ineffective [17,18], posing challenges which this paper attempts to address. Under some conditions, co-management combining action by government and user organizations may be effective but also shows many limitations, including government capacity and political economy [19]. Much of the literature on groundwater governance offers ideas that are most relevant where there is sufficient administrative and technical capability as well as political agreement to impose regulations [20]. However, such favourable conditions are often not present. This paper is particularly concerned with ways to improve groundwater governance under conditions with limitations in information, administrative capacity, resources, and political support for groundwater governance.

Conceptualization of the development of groundwater governance in "stages" may lead to an expectation that wealthy industrialized countries offer the most relevant models for groundwater governance, with other places assessed in terms of their "gaps" and "deficiencies" [21–23]. Oversimplified conceptions of governance may lead to an expectation that, for example, India and China must eventually end up governing groundwater much like California, Spain, or Australia. Analysis in terms of unilineal conceptions of modernization may also lead to a (mis)conception that development of groundwater governance requires first establishing comprehensive and integrated institutions incorporating best principles and practices for good governance. This may result in prioritizing the construction of formal institutions, including laws, policies, plans, and administrative agencies and organizations, which may yield the appearance of regulation, even if these are largely ineffective, a "paper state" [24,25].

On the other hand, the synthesis of experience with groundwater development in South Asia offered by Shah [12] suggests that groundwater depletion may be hard to avoid, similar to the history of overexploitation of other renewable resources such as forests and fisheries. Severe aquifer depletion has occurred not only in many parts of India, where there is significant state capacity compared with many other developing countries, but also in the North China Plain with an even stronger state, and in areas in the United States such as the high plains and central California. These examples indicate the difficulty of achieving political agreement on groundwater regulation, even where there is substantial information and technical and administrative capacity.

Efforts to regulate groundwater and other common-pool resources can be susceptible to capture or obstruction by narrow interests [26]. From an institutional and resource economics perspective, the potential for groundwater governance may be affected by potential benefits and costs, including transaction costs; the ability to observe groundwater use, impacts, and boundaries; number and heterogeneity of users; and the complexity and availability of information about aquifers, recharge, and other stocks and flows [27,28]. Rather than assuming that more governance necessarily means better governance, development of governance institutions may better be seen in terms of strategically optimizing expected gains and costs for groundwater users and others, including the transaction costs of institutional change.

As mentioned in the introduction, availability of groundwater, falling costs for solar pumping, and development of favourable conditions for other factors, such as growth in road networks and urban and international markets for high-value horticultural crops, suggest the potential for a groundwater boom in Africa. This might grow largely out of government control, and perhaps despite any formal regulatory frameworks nominally in place. The potential for such expansion makes it all the more relevant to consider ways in which water users and their organizations might under some conditions be willing to work together to govern groundwater, on their own or in conjunction with governments.

### 1.4. Polycentric Solutions and Institutional Artisanship

There is increasing interest in locally based approaches to groundwater governance, in part due to pessimism about the feasibility of centralized regulation. Experience in Africa and elsewhere does reveal some examples of people and organizations working together to solve problems related to groundwater. Crafting institutions through which people and organizations cooperate to govern groundwater as a shared resource can be seen as "institutional artisanship" [16,29,30].

However, there are also concerns and critiques about the inadequacies of community-based approaches to natural resources management, particularly external efforts to induce participation, suggesting the limitations of past policies and the need for different approaches [31,32]. Local governance may not necessarily emerge, despite worsening problems. Much recent expansion of irrigation in Africa appears to occur on an individualized basis, with informal land tenure and without necessarily having effective communal institutions available for regulating resource use or resolving conflicts, i.e., not necessarily as part of a "commons" that can be assumed to already be well-governed [13].

Collective action to govern groundwater lies between atomized, uncoordinated individual use and hierarchical government control. Cooperation to govern groundwater can be seen as a form of polycentric governance [16,33]. Polycentric governance involves multiple autonomous organizations interacting through mutual adjustment including competition, conflict, and cooperation [34–37]. Polycentric systems can be seen as heterarchies with distributed power [38], in contrast to hierarchies with centralized power and chains of command or decentralized networks with power fragmented or dispersed at the lowest level. Polycentric governance offers opportunities for citizens to craft institutions to serve their needs. As Elinor Ostrom stated, "[b]y 'polycentric' I mean a system where citizens are able to organize not just one but multiple governing authorities, as well as private arrangements, at different scales" [39].

Polycentric governance can include not only distributed community-level efforts, but also nested, multi-level, and cross-scale linkages between organizations [16,33]. Polycentric governance institutions may be constituted through agreements between organizations such as irrigator's associations and municipal water providers, within the context of overarching authority from a larger government that allows internal autonomy. Groundwater governance may be polycentric in the simple sense of having multiple centres, not under hierarchical control, and perhaps learning from each other. Polycentric governance may also involve peer-to-peer cooperation or interaction across smaller and larger scales.

This paper looks at various ways people have been able to cooperate to improve use of groundwater, particularly at the kinds of things that farmers and water management organizations may be able to do largely on their own initiative or with limited government support. This paper is based on an initial desk review of selected sources from available literature, particularly for groundwater use in sub-Saharan Africa. The paper explores some institutional options for polycentric groundwater governance that deserve further attention.

The goal of this paper is a better appreciation of voluntary or self-governance opportunities, how they might work, and how they might be improved. This can contribute to better policies and to better understanding of the diversity of water governance institutions and their dynamics. This does not rule out state-centred or state-led groundwater governance but is intended to offer insights into a diverse range of possibilities. Polycentric governance can expand the potential for governance solutions but is not a panacea [40]. There may be many conditions under which collective action and polycentric efforts to govern groundwater are absent, fail, or are only partially successful. The question here is to better recognize and understand what forms successful collective action may take, and what actions may encourage the emergence of better groundwater governance.

### 1.5. Structure and Synopsis of the Paper

The next section of the paper presents an extended ladder of participation [41,42] as one way to analyse cooperation in groundwater governance, particularly relationships between government agencies and collective action by water users. Drawing on the literature on groundwater governance in Africa and relevant experience elsewhere, the Section 3 presents findings about different kinds of cooperation that have sometimes achieved improved outcomes in groundwater governance, discussing experience internationally and in Africa. These offer examples and "existence proofs" for forms of cooperation that have been feasible in some cases outside of state management or state-led co-management. This includes but is not limited to purely community-based or farmer-led forms of groundwater governance. Such alternatives include:

- the use of simple local rules, such as enforcing spacing between wells;
- sharing of wells, pumps, and pipe networks through joint investment or water sales;
- protection of domestic water sources against impacts from increased pumping;
- watershed land improvement to harvest rainwater for storage in aquifers;
- coordinating crop types, areas, and timing to fit groundwater availability and sustainability; and
- cooperation to import water to replenish and manage groundwater reserves.

The Section 4 discusses ways in which resource users, non-government organizations, and governments may be able to promote more effective governance of groundwater, including providing information, funding research, facilitating association, enabling self-governance, endorsing norms and standards, and subsidizing investments, as well as delegating authority or establishing empowering legal frameworks. The final section briefly summarizes conclusions and recommendations.

## 2. An Extended Ladder of Participation

In their recent review of groundwater co-management, Molle and Closas [19] arrange an interesting set of cases roughly along a spectrum of greater or lesser government control. However, they claim that there is no "metric" for co-management. One way to provide a useful metric and understand stakeholder relationships is in terms of a ladder of participation [41]. This can be based on the extent to which one organization shares information, listens, discusses, collaborates, or makes decisions jointly with another. An extended ladder of participation [42] as shown in Table 1 can go beyond joint management to cover situations where non-state organizations play the primary role, but may still cooperate, receive advice, share information, or benefit from an enabling legal framework. Levels of participation represent a way to provide a simple summary of a complex set of

relationships, which could also be analysed in more detail in terms of different kinds of social network linkages and associated forms of power and influence.

**Table 1.** An extended ladder of participation.

| | |
|---|---|
| **FARMER-LED/ COMMUNITY-BASED** | 9. **Enable**: Framework providing legal status and recourse for organizations and individuals. Accountability based on organizational charter, general reporting and auditing requirements, property rights, contracts, liability, etc.<br>8. **Advise**: Provision of information, guidance, and other technical assistance as input to decisions, guidance for voluntary compliance, e.g., extension, statistical information and research, promotion of voluntary initiatives and coordination, technical standards<br>7. **Establish autonomy**: Institutions constituted by users such as through court-sanctioned dispute settlement, corporate structure for shared ownership, or special districts based on enabling legislation. Autonomous decisions by communities, organizations (associations, companies, etc.), or individuals, subject to compliance with specific laws and regulations, e.g., regulatory review for issuance and renewal of permits and licenses, recognition of customary rules and practices, enforcement of general environmental standards |
| **JOINT/CO-MANAGEMENT** | 6. **Delegate authority**: Decisions by a group or organization empowered with specific authorization, e.g., devolution by legal mandate, management concession, operating franchise, commission-delegated power for final decision, etc. especially to the extent that authority is revocable or subject to approval of plans, budgets, and expenditures; detailed supervision; or override<br>5. **Partner**: Joint decisions by mutual agreement, consensus, co-operation where both sides hold veto power, e.g., some co-management agreements, intergovernmental organizations, public–private partnerships, contracts<br>4. **Collaborate**: Stakeholder representatives "at the table", active as team members in formulating and recommending alternatives, although final decision by one party. Task forces, working groups, negotiated rulemaking |
| **STATE-LED** | 3. **Involve**: Interactive discussion and dialogue, as a supplement to an existing internal decision process. Workshops, town hall meetings, charettes, some advisory groups<br>2. **Consult**: Two-way communications, receiving input, listening, exchange of views. Public hearings, written comments, question & answer sessions, interviews, focus groups, questionnaire surveys, etc.<br>1. **Inform**: One-way information dissemination about problems, analysis of alternatives, plans, and decisions, e.g., announcements, lectures, brochures, press releases, press releases, websites, reports, etc. |

Source: Adapted from Bruns (2003).

In contrast to the frequent emphasis in the literature on top-down regulation, co-management, or purely local-level action, this paper is mainly concerned with voluntary governance with various horizontal or vertical linkages. This concerns situations where non-state actors can play primary roles, interacting with other organizations, sometimes including government agencies. Voluntary governance typically occurs within the overarching context of larger government jurisdictions. In contrast to a state-centric perspective on how stakeholders can "participate" in government-led policies and projects with external enforcement of regulations, the main concern here is with how governments might "participate" in groundwater governance where others take primary roles.

## 3. Participatory and Polycentric Alternatives

Experience in India, Africa, and elsewhere demonstrates a range of institutional options for collective action to govern groundwater, including through locally initiated cooperation as described in this section. These examples illustrate the potential for farmer-led and community-based activities in the extended ladder of participation. Such activities are often oriented towards solving specific problems or realizing particular opportunities. Such action illustrates possibilities that may be pragmatically and politically feasible, and serve as a pathway for further institutional development.

Awareness of potential institutional options can help in identifying possibilities that might be relevant in particular cases, as well as suggesting the potential for innovative adaptations to local circumstances. Analysis can offer some insights into actions that influence the feasibility of various governance actions, particularly for situations where there is limited potential for top-down regulation or for resource-intensive forms of co-management. For planning and policy-making, a polycentric perspective may help formulate a better "theory of change" about possible actions and results, including ways to efficiently target action to improve governance, reduce risks, and monitor results [43]. Even in contexts where governance at least in principle takes the form of centralized regulation or state-led co-management, it can still be important to understand how and why stakeholders may be more likely to engage, take initiative, resist, or selectively utilize aspects of a formal regulatory framework.

### 3.1. Local Rules

Traditionally, some communities have applied simple rules, such as requiring a minimum spacing between wells [44]. This may be driven by fear that new wells, especially deeper tubewells with motorized pumps, could dry up neighbouring wells. This could include temporary interference (cone of depression due to pumping) or permanent lowering of the water table due to cumulative groundwater extraction. Installation of wells is usually relatively easy to observe, and, if there is sufficient local consensus or authority, it may be feasible to stop construction or even close installed wells. Examples from Yemen illustrate how such local conflicts and governance over well installation may play out even in the context of very limited central government authority [45]. Even if not completely successful, consideration, contestation, or agreement about such rules may encourage more deliberation and coordination about changes that will have major impacts on neighbouring farmers and so influence how well installation and groundwater extraction proceeds.

In Africa, as elsewhere in the world, there is long experience with wells, springs, and wetlands, which often includes customary law about sharing access to water, priorities for different uses of water, protecting water quality, and other matters [46]. These form a cultural context within which people observe and learn about new possibilities, including drilling deeper, using motorized pumps, and powering pumps. In areas with Islamic influence, customary concepts may include the "right of thirst" and other teachings about sharing water and delimiting zones to protect water sources. There may already be ideas about avoiding harm to others and customary tenure regulating access to land and other natural resources. Localities and legal systems differ in the extent to which communities or local authorities provide recourse and redress to those harmed by violations of customary rules. As new technologies offer new opportunities and challenges, these are likely to still be viewed through the lenses of existing ideas such as concepts concerning water, access, and purity. Rather than simple erasure and replacement of older ideas, change may well result in a blend or institutional bricolage of ideas, which shape local conceptions and contestation about priorities and possibilities [47].

### 3.2. Sharing Wells, Pumps, and Pipes

Neighbouring farmers may share a well, and possibly also pumps and pipe networks, taking advantage of economies of scale. This can occur through joint investment, such as by siblings, relatives, or neighbours, especially if deeper wells and pumps require more capital. In the case of Yemeni farmers irrigating a high-value crop (qat), such joint investment has financed wells hundreds of meters deep, costing tens of thousands of US dollars, and sometimes includes sharing pipe networks for water distribution and linking multiple wells to diversify sources and reduce risks [45]. Agreements about sharing investments and sharing water can constitute a form of private governance [48,49]. As with customary rules, private agreements provide another form of governance as an alternative or in addition to state-promulgated rules and regulations. Such collective action may accelerate irrigation development and may also accelerate aquifer depletion. Such sharing of infrastructure can

also provide a context within which water users respond to declining water levels, threats to domestic water supply sources, and opportunities for increasing recharge. The same channels for communication, negotiation, agreement, and cooperation may play a role in attempts to make groundwater use more equitable, efficient, or sustainable.

At an aquifer level, shifts to make joint investments and share wells, pumps, and linked distribution networks can be seen as steps toward developing institutions to govern groundwater and aquifers as common property. Conceptually, this could also be seen as incremental steps toward partial "unitization" of a common pool resource. In the oil and gas industry in the western United States, larger-scale cooperation has been supported by legislation for unified management of oil fields by a single operator or coordinated operation of wells under shared rules [26,50]. Ideally, extraction and other activities can be managed in ways that are more efficient and avoid damage from a "race to the pump" including conflicts between neighbouring users and compaction that permanently loses storage capacity in the aquifer.

Farmers may use wells and pumps to deliver water to others for payment. On a wider scale this can create local markets for irrigation water supply, as in Pakistan and elsewhere [51]. In some such areas, water from surface irrigation systems replenishes groundwater, creating the potential for conjunctive use and management of surface water and groundwater. While government projects have also tried to set up arrangements for shared wells and pumps, the sustainability of such externally organized efforts can be problematic, especially for maintenance and replacement of expensive equipment. In trying to understand current practices and the opportunities for polycentric governance of groundwater, it is relevant to look at the extent to which there is some degree of sharing of wells, pumps, and pipes as part of an array of institutional options.

### 3.3. Prioritizing Drinking and Domestic Use

Communities may sometimes block well construction, enforce well spacing, and take other measures to stop projects that they fear would endanger wells, springs, and streams that they rely on for domestic water supply, as occurs for example in Rajasthan State in India [52]. This reflects a willingness to prioritize some water uses over others. Sometimes, irrigation use may be allowed as long as water is abundant, but irrigation restricted and water reserved for use by humans and livestock as water becomes scarcer during the dry season or in times of drought. Such restrictions may seem obvious or common sense, but it is worth noting the extent to which they do occur, as well as whether and why such efforts may be effective or ineffective. In formal groundwater governance, some aspects of this may be discussed in terms of "protection zones" a concept which can also be seen in traditional Islamic Law [22].

In many parts of Africa, there is a long history of relationships between pastoralists and farmers regarding access to water and grazing, which can be an ongoing source of co-operation and of tension and sometimes severe conflict. Reductions in the cost of pumping using solar power, perhaps in combination with improved access to drilling technologies and markets for farm inputs and outputs, occur within such already contested contexts.

### 3.4. Recharging Groundwater

Mass movements to harvest rainwater and replenish aquifers, through building ponds, berms, trenches, and other earthworks to capture rainfall run-off and increase infiltration into aquifers, have been a prominent response to increasing water scarcity in parts of India [12]. Some of these have been locally initiated, and some have received support from religious leaders, non-government organizations, and government agencies. Perhaps counter-intuitively, this appears more likely to occur in hardrock aquifers that have very limited storage capacity and are easily depleted, but sometimes may also be easily replenished [4]. In such aquifers, not only do farmers experience shortage more quickly, but they can also more easily observe the results of action to recharge groundwater.

Like India, much of Africa, estimated at around 40%, is underlain by weathered hardrock geology, such as granites and basalt, with relatively shallow weathered regolith aquifers that have limited storage capacity [53–55]. These may be easily depleted but could also be deliberately recharged, and so may offer more favourable conditions for collective action than alluvial aquifers. In addition to deliberate efforts to recharge aquifers, which are still less frequent in Africa [56], groundwater recharge may be increased by efforts directed at improving watershed conservation and rainfall retention, such as through construction of small reservoirs, sand dams in riverbeds, on-farm water storage in ponds, trenches, and earthen or rock bunds between fields. Lack of information about aquifers, high heterogeneity in aquifer conditions, and changes over time put a premium on polycentric approaches that can build on available local knowledge and adaptive learning to make better decisions and learn from experience.

Some areas in the Sahel have seen widespread implementation of measures to retain rainfall run-off and reduce erosion, which can contribute to increasing groundwater recharge. Construction of small reservoirs and ponds may also influence hydrology, locally and downstream, by affecting recharge, evaporation, and irrigation water use. Alluvial aquifers in floodplains often receive annual recharge as rivers rise during a rainy season. Recession agriculture as river levels fall is widely practiced in some areas, such as along the Niger River in West Africa. Pumps may facilitate recession cultivation. However, surface and groundwater along rivers may also be strongly affected by dam construction and increased water consumption upstream.

### 3.5. Coordinating Crops

Farmers may act together to control water demand by agreeing about crops to grow, and when and how to grow them. Concepts for top-down regulation of groundwater typically assume the feasibility and necessity of controlling how much water is withdrawn, by licensing wells, installing flow meters, and imposing volumetric limits on abstraction. However, groundwater abstraction by many dispersed users is usually hard to monitor and even harder to control. Planting of crops is much more visible and decisions about crop choice easier to influence. Availability of remote sensing imagery such as from satellites and drones opens up more opportunities to monitor crop cultivation and evapotranspiration, observing cropping patterns and estimating water consumption without requiring flow meters on individual wells or direct access to fields.

Projects in Andhra Pradesh have shown that, at least under some conditions, coordination could be developed primarily through an "information-based" approach. Participatory hydrological monitoring and community coordination on crop-water budgeting could occur through voluntary action concerning crop choices, such as switching from irrigated rice to tomatoes or other higher-value horticulture, without necessarily requiring rules with sanctions against violators [57–59]. Sustainability of the formal organizations developed by the project seems problematic [60]. However (based on the author's field observations together with colleagues from the Foundation for Ecological Security), farmers visited at former project sites said they did continue with improved local understanding of groundwater, hydrological observation and consideration in well-siting and well-deepening, and coordination on water-thrifty farming practices. For such voluntary change, it may be particularly important to have a profitable alternative crop that uses less water, and a critical mass of farmers who agree to conserve. In other cases, coordination of crop choices could depend on rules with enforcement through social pressures or other sanctions.

### 3.6. Importing Water and Managing Aquifer Reserves

In southern California, irrigation districts and water utilities have joined together successfully in some cases to reverse saltwater intrusion and replenish aquifers so they could provide reserves for times of shortage [16,33,61]. When conflicts are taken to court, settlement agreements under court authority can provide a framework for cooperation to resolve disputes. In response to recent droughts, the State of California has strengthened

the legal framework enabling and requiring groundwater users in local areas to devise plans to manage groundwater more effectively [62–64].

Typically, it is much easier to mobilize collective action to lobby for increasing the supply of water or subsidies, than to control demand for water [12,33]. Efforts to import water may incorporate limitations or moratoria (freezes) on expansion of crops and wells. Such cooperation may then become a basis for additional forms of management, such as reserving some stored groundwater for periods of drought. However, results from top-down project plans for surface water imports to facilitate changes in groundwater management may turn out differently than expected, as occurred in Morocco [65].

## 4. Implications for Supporting Participatory and Polycentric Solutions

As illustrated by cases such as those presented above, there are a variety of ways in which governments, non-government organizations, farmer groups, and others could support the growth of polycentric problem solving in groundwater governance, which can be seen as examples of state-supported self-governance [16,66].

### 4.1. Information for Mutual Understanding

Better information can improve common knowledge and mutual understanding of what is happening with groundwater and what might be done about it. In the context of making rules about well spacing, protecting drinking water sources, or recharging groundwater, it may be particularly useful to have a shared understanding of aquifer characteristics such as recharge and discharge areas, storage capacity, and the direction and speed of groundwater movement, since these can be quite heterogenous and sometimes hard to predict. Here, external experts and scientific knowledge may be able to contribute, especially if expert advice is provided in a way that can be combined with local knowledge, as occurs, for example, in integrated pest management and some forms of participatory rapid appraisal (PRA).

Information can be made available using a variety of communication technologies including older methods such as meetings, study tours, leaflets, radio, and television as well as newer technologies such as the internet, websites, and smartphone apps. These are becoming increasingly widespread even among poorer farmers in Africa, albeit still with differences in access affected by poverty, gender, and other factors. Information can be customized to fit detailed local conditions, for example using geographic information systems (GIS) to aggregate, analyse, and display relevant information. Research can produce a better understanding, particularly of things that are hard to observe directly such as a conceptual understanding of aquifer structure, trends over wider areas, estimates of the volume of water in aquifers, and groundwater movement, as well as water quality.

Activities for improving information can include citizen science [67,68] such as participatory monitoring of groundwater levels and rainfall, education regarding groundwater recharge and discharge patterns, and participatory rapid appraisal of local experience with wells. Experimental games simulating different crops and aquifer depletion can aid understanding about relationships and potential for collective action to govern groundwater [69]. The process of gathering such information may lead to changes in perceptions and values, i.e., changes in environmentality [70], as well as the creation of common knowledge and interpersonal relationships.

Water quality problems, such as arsenic, are not immediately observable and are an example of where government may play a strategic role in water governance [71]. Information about the presence of arsenic in groundwater in Africa is limited [72]. Occurrence of arsenic appears to be heterogenous and highly localized. Hazard maps based on geological information can help identify areas where arsenic is a risk [73,74]. However, presence of arsenic often varies from one well to the next even within local areas, so water testing of specific wells is crucial. Test results available to water users can support informed decision making. There are options for treatment to remove arsenic; however, these require a degree of technical skills, operating capacity, and continuing financing, which often may not be

feasible or sustainable. In many cases, it may however be feasible to switch to a different well or other safer source for drinking water, which may be a practical option that could occur through informed community action.

### 4.2. Facilitating Trust and Cooperation

Bringing stakeholders together can help them learn from each other, create common understanding of groundwater issues, and develop relationships and craft solutions. As mentioned above, activities to improve information may also contribute to developing relationships and discussion about problem solving, as with groundwater games in Andhra Pradesh [69]. Community organizers may facilitate development of collective action. Multi-stakeholder platforms can bring together diverse participants concerned with irrigation, domestic water supply, industry, environment, and other issues [75–80]. Appreciative Inquiry [81] and other facilitation methods can foster engagement in polycentric problem solving [82]. Multi-stakeholder approaches can bring together a broader range of perspectives and concerns and thereby reduce the risk of governance being captured by narrow interests. The success and survival of such efforts may depend crucially on finding feasible ways to go beyond just talking and instead act together to govern groundwater, starting with easier or less controversial actions or just joint fact finding. Such trust building can apply for joint efforts with government agencies, as well as for autonomous local action, perhaps supported by information and empowering legislation.

### 4.3. Norms and Standards

Standards can embody the results of analysis and experience regarding how to assess and manage groundwater. They can identify what is considered desirable and feasible, for example in well siting and construction. These could be based on voluntary choice about whether to use particular standards. They could concern technical efficiency for equipment, environmental stewardship, and other topics. Standards and certification could also be required by regulators or by others such as insurance companies, banks, or companies selling for particular markets. It is important to note that in many cases formal laws and regulations may function largely by providing guidance about acceptable standards, in which most participants are motivated by voluntary compliance, even if sanctions may matter for some.

Even when standards are legally mandated, the way in which they function may largely be a matter of voluntary response to available information. In their analysis of arsenic in groundwater in Ghana, Tanzania, and Bangladesh, Nwankwo et al. [83] say: "the adoption of a national standard is effectively voluntary, as compliance is not enforced. It has limited statutory value as in most cases a consumer cannot make anybody liable for breach of standard limits . . . ". Availability of water testing results and appropriate standards could enable informed decisions about using alternative supplies and water treatment, illustrating one way in which government (or private) testing services could facilitate individual and collective action to obtain safe water.

### 4.4. Empowerment

At the top level of the participation ladder, government legal frameworks can enable people to form associations and enter into agreements, including articles of association, contracts, and settlements to resolve legal disputes. Laws and regulations may also provide a basis for rights, including rights to water that could be defended in court and may authorize taking action against those who cause harm. Legal systems may also recognize existing customary arrangements or enable formation of new organizations with formal authority to make rules about resource use, mobilize resources, make repairs, and improve infrastructure. Typically, governments often try to promote specific arrangements for water user organizations, with formal requirements and often a high degree of top-down control, supervision, or tutelage, limiting autonomy. However, rather than restricting legal support to a single organizational form highly dependent on a particular agency, regulatory

frameworks could also allow more flexibility including acceptance of existing customary governance, and roles for multiple different kinds of organizations, single-purpose and multi-purpose, including local governments.

A recent review of water user associations (WUAs) in sub-Saharan Africa by International Water Management Institute researchers [84] concluded that approaches to developing standard "mainstream" water user associations had generally failed to develop effective and sustainable organizations. This includes efforts directed at reforming national laws and regulations, formulating local bylaws, technical and administrative training to build institutional capacity, management transfer to WUAs, and contracts between WUAs and government agencies. The review suggested that there was little reason to expect better results from efforts at increasing private sector roles in service delivery, such as public–private partnerships. The main recommendation was for increased state support with continuing budget and staff to work with WUAs, including facilitating inclusion and equity and improving monitoring and evaluation.

The review identified a range of "alternative management options" that might be relevant in specific contexts including participatory design; joint management for water service delivery; multi-stakeholder processes for innovation in value chains; multi-functional WUA services for agricultural inputs and marketing, including finance; participation in basin water allocation; and WUAs combining responsibility for surface and groundwater in a command area. These alternatives could be done within the context of current WUA policies and programmes while allowing a clearer focus on particular irrigation-related activities (tasks/functions) and results.

Many of these alternatives could also be pursued directly, leaving organizational arrangements more flexible or optional without necessarily requiring a particular organizational model or assuming or requiring that a formal organization must be established first. For groundwater, this could include "information-based" approaches that emphasize extension, education, and participatory hydrological monitoring to create changes in understanding and local consensus; crop-water budgeting and other approaches to improving coordination in cropping decisions; and watershed management activities to increase recharge. Conceptually, such approaches could be seen as emphasizing networks of information and influence, including extension approaches such as contact farmers and farmer-field schools.

International experience, current and historical, includes a variety of institutional alternatives beyond conventional WUA models. Local governments may manage irrigation directly, as done earlier in parts of Indonesia and often still perpetuated in various forms, or through cooperatives or other bodies under the auspices of local governments as in Vietnam, Turkey, and elsewhere. Local governments may also support arrangements organized around individual roles for customary watermasters or service provision by specialized individual contractors.

Many efforts to develop WUAs provide little or no power to WUAs, for example to enforce fee collection or compliance with rules. By contrast, "special districts" in Japan, USA, Spain, and elsewhere are dedicated forms of local government, with procedures for establishment and oversight and strong authority to levy fees and enforce rules regarding infrastructure and water use. Collective action in irrigation can also be legally structured as corporations, for example as non-profits, mutual companies, or social enterprises.

Some of the accounts of emerging or farmer-led irrigation development also appear to include informal but longer-term arrangements for access to land. In contrast to short-term seasonal rental or sharecropping, these include investments in wells, pipes, and land improvement. In terms of resource tenure, these appear more like leasing, with various mixes of responsibilities between owners and leaseholders. The importance of capital investment and technology in some forms of groundwater irrigation such as centre-pivot systems suggests the potential relevance of institutional arrangements analogous to franchising models, with somewhat standardized packages of technical and financial components combined with local investment and operation.

Kenya's water policies have shifted over time to support more polycentric governance, with a more powerful role for local organizations, though mainly for surface water [85]. Polycentric governance can encourage a broader range of innovation, experimentation, and learning. However, it is certainly not perfect nor a panacea. Looking at two cases of irrigators growing horticultural crops near Nairobi, Kanyua [86] found that efforts to develop local water governance imposed top-down preconceptions about how to organize, focused too narrowly on surface water allocation, omitted groundwater usage, and neglected the social capital available in existing marketing groups, resulting in "imposed self-governance" with limited effectiveness.

Thus, a broad range of legal institutions could empower collective action in irrigation, including cooperation and support from government well beyond conventional WUA models. Even within current conceptions of WUAs in water governance, there are many options for targeting efforts on specific activities and results, including better-informed participatory planning, field-level water service delivery, and linkage with input and output markets (value chains).

### 4.5. Subsidies

Governments may respond to water shortage and drought by funding investments, for example in wells and pumps [12]. Under the pressure of drought and concern about impacts of climate change, irrigation and solar pumping are likely to offer promising targets for aid. Farmers may lobby based on a shared interest in obtaining subsidies while governments and politicians see subsidies as useful to show action and obtain political support. However, rather than directly subsidizing the cost of equipment, it might be more effective to introduce examples of easily available irrigation services, so that farmers can experiment, learn, and take the initiative for further expansion [1].

In the case of groundwater and concern about shortage, there are also risks that subsidies may lead to unintended consequences that make conditions worse rather than better, increasing water consumption (evapotranspiration) and reducing return flows. This makes it important to look carefully at the net impacts of measures intended to improve irrigation "efficiency" [43,87,88]. Subsidies could be better designed to support worthwhile activities such as watershed conservation and groundwater recharge using stormflows, thereby minimizing negative impacts downstream and instead contributing to increased storage and baseflows. Subsidies can also be designed and analysed according to the extent to which they allow or encourage local autonomy.

### 4.6. Regulation

This paper focuses on the potential for action primarily by stakeholders working together in improving management of groundwater. This may involve water users in various ways, including joint activities for co-management, as well as ways in which regulations may provide a clear framework for private action, to reduce uncertainties and provide ways of resolving disputes through courts, mediation by local authorities, and other means. Licensing of wells and detailed control over water abstraction may be relevant and feasible in some cases [17,89] but are far from the only option for government action regarding groundwater governance. Even where regulation is present, local actors often have great scope in the extent to which they do or do not follow rules and support their implementation.

Hybrid regulatory approaches offer a strategy to focus licensing and other control on large, high-impact users while allowing smallholders more opportunity to irrigate and improve their incomes [90]. Regulation may become more urgent and acceptable after initial phases of exploration and expansion in groundwater use and as problems become more apparent [14]. In addition to the potential gains from cooperation, local agreement to establish new forms of authority and enforceable sanctions may be motivated by the attempt avoid adverse court rulings or political dictates. Cooperation and compliance may be motivated not just by financial incentives or sanctions, but also moral considerations,

legitimacy, participation, and voluntary acceptance by many of those involved. This represents another way in which voluntary governance may still play a role even within the context of hierarchical power and rules with effective sanctions.

*4.7. Polycentric Discovery*

A polycentric process can help identify opportunities for improving groundwater governance through voluntary governance and institutional artisanship. Polycentric governance adapts institutional solutions to local needs and circumstances, within overarching institutions that can encourage attention to wider environmental impacts and social equity. Exchange of ideas among diverse experiments accelerates social learning. Polycentric governance is not a panacea but can expand opportunities for people to craft meaningful solutions for improving their lives, including how they irrigate and how they manage aquifers. In terms of policy and strategy, emphasis on social learning and institutional innovation could be supported by approaches emphasizing polycentric problem solving, including for example, problem-driven innovation and adaptation (PDIA) [24] and adaptive management that would be open to recognizing and learning from a diverse variety of institutional arrangements.

This paper offers an exploratory analysis of opportunities for polycentric groundwater governance in Africa, which could be further developed in a variety of ways. An extended ladder of participation provides one fairly simple lens for examining institutional opportunities beyond state-led regulation and co-management. Understanding of these opportunities could be expanded by looking at more cases of what groundwater users in Africa are already doing collectively as they increase utilization of wells and pumps for irrigation. Analysis could be deepened by examining the socio-ecological systems involved in groundwater governance, including physical, social, and economic characteristics such as size, heterogeneity, and the potential gains, losses, and transaction costs that affect institutional change. Ethnographic and appreciative approaches can enhance understanding of how those involved perceive their situations, choices, and outcomes. Elinor Ostrom's institutional design principles [16,91,92] focus attention on crucial institutions for effective governance and how resource users and other stakeholders can craft governance to fit their circumstances and goals.

## 5. Conclusions: Opportunities for Polycentric Solutions

Analysis of experience in Africa and internationally shows the potential for a diverse range of ways in which people and organizations may work together to govern groundwater. Atomistic individual action and top-down licensing of wells with control of water withdrawals are far from the only institutional options for groundwater governance. Table 2 summarizes some differences between conventional models for top-down regulation of groundwater and polycentric approaches with more local autonomy. Such activities offer a broader array of options for institutional artisanship, expanding the solution space for institutional design and facilitating the discovery of ways to improve governance. The ideas presented here offer alternatives for a diverse mix of institutional solutions, on their own or in combination with co-management and government regulation. Local efforts to govern groundwater can take particular forms that make management effective, such as simple rules about well spacing, controlling demand by coordinating highly visible decisions about crops, prioritizing and protecting domestic water sources, harvesting rainfall to replenish groundwater, and actively managing the storage capacity in aquifers. The opportunities for collective action in groundwater governance can be enhanced by activities that improve information, facilitate association, endorse standards, and provide a legal framework for agreements.

**Table 2.** Alternatives for polycentric water governance.

| Conventional Control | Alternatives |
|---|---|
| License well installation | Simple rules, e.g., well spacing |
| Install meters to measure abstraction | Participatory monitoring of rainfall, groundwater levels, and cropping |
| Control water withdrawal | Coordinate crop choices and timing |
| Enforce rules with penalties | Craft consensus on win-win strategies |
| Limit water use for irrigation | Prioritize domestic use |
| Apply best practices | Improvise and adapt management to fit complex local conditions |
| Stop depletion | Replenish aquifers |

**Funding:** Not applicable. There was no funding for this study.

**Institutional Review Board Statement:** Not applicable. The author has no institutional affiliation.

**Informed Consent Statement:** Not applicable. There were no interviews or other use of human subjects.

**Data Availability Statement:** Not applicable. There is no data set beyond the references cited in the article.

**Acknowledgments:** An earlier version of this paper was presented at the Conference on Voluntary Governance "Artisanship in Culture and Enterprise" 5–7 November 2020, Arizona State University, Tempe, AZ. Content and views expressed in this paper are the sole responsibility of the author and not of any organization with which he is or has been affiliated. Some ideas in this paper draw on the author's consulting work with colleagues in the Water Commons Project at the Foundation for Ecological Security in India. The paper has also benefited from discussions with Jude Cobbing. The author is responsible for errors and omissions.

**Conflicts of Interest:** The author declares no conflict of interest.

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
