# Peer review of "Polycentric Solutions for Groundwater Governance in Sub-Saharan Africa: Encouraging Institutional Artisanship in an Extended Ladder of Participation"

_water, doi:10.3390/w13050630_

Round 1

Reviewer 1 Report

The paper titled “Polycentric Solutions for Groundwater Governance in Sub-Saharan Africa: Encouraging Institutional Artisanship in an Ex-tended Ladder of Participation”, presents a good topic for readers of this Journal. Hovewer, several some lacks emerge after reading the paper. Below is the list of serious lacks to justify my decision to reject the manuscript.

  • Lack of novelty. This manuscript represents a poor data analysis.
  • Lack of an adequate method description.
  • absence of figures representative of the results.

Reading the title I expect to find a DSS useful for the groundwater management in Africa. For next submission I suggest to consider following papers:

  • 1- Molina, J.L., Bromley, J., García-Aróstegui, J.L., Sullivan, C., Benavente, J., 2010.Integrated water resources management of overexploited hydrogeological systems using object-oriented Bayesian Networks. Environ. Modell. Softw. 25 (4),383–397.
    2- Al-Senafy, M., Abraham, J., 2004. Vulnerability of groundwater resources from agri-cultural activities in southern Kuwait. Agric. Water Manage. 64, 1–15.
    3- Voudouris, K., Polemio, M., Kazakis, N., Sifaleras, A., 2010. An agricultural decision support system for optimal land use regarding groundwater vulnerability. Int.J. Inf. Syst. Soc. Change 1 (4), 66–79.
    4- Henriksen, H.J., Rasmussen, P., Brandt, G., von Bulow, D., Jensen, F.V., 2007. Par-ticipatory modelling using Bayesian networks in management of groundwater contamination. Environ. Modell. Softw. 22 (8), 1101–1113.
    5- Knuppe, K., Pahl-Wostl, C., 2011. A framework for the analysis of governance structures applying to groundwater resources and the requirements for the sus-tainable management of associated ecosystem services. Water Resour. Manage.25, 3387–3411.
    6- Stefanopoulos, K., Yang, H., Gemitzi, A., Tsagarakis, K.P., 2014. Application of themulti-attribute value theory for engaging stakeholders in groundwater protection in the Vosvozis catchment in Greece. Sci. Total Environ. 470–471,26–33.
    7- Van Camp, M., Radfar, M., Walraevens, K., 2010. Assessment of groundwater stor-age depletion by overexploitation using simple indicators in an irrigated closed aquifer basin in Iran. Agric. Water Manage. 97, 1876–1886.

Author Response

The paper titled “Polycentric Solutions for Groundwater Governance in Sub-Saharan Africa: Encouraging Institutional Artisanship in an Ex-tended Ladder of Participation”, presents a good topic for readers of this Journal. Hovewer, several some lacks emerge after reading the paper. Below is the list of serious lacks to justify my decision to reject the manuscript.

  • Lack of novelty. This manuscript represents a poor data analysis.
    • The paper responds to cited recent review articles by Molle and Closas summarizing problems of current approaches to groundwater governance, and as such also addresses the limitations of the approaches presented in synthesis work by Villholth et al. in Advances in Groundwater Governance. It offers a novel presentation of approaches relevant for applying polycentric governance, particularly for conditions where information, experience, and institutional capacity are limited. 
  • Lack of an adequate method description.
    • Text has been modified to further clarify that the paper is based on an initial desk review of available literature. As stated in the paper, this provides cases that offer examples and in some cases "existence proofs" of what may be feasible under some conditions. The conceptual framework of an extended "ladder of participation" is clearly presented in the paper. 
  • absence of figures representative of the results.
    • Figure 2 summarizes key results.

Reading the title I expect to find a DSS useful for the groundwater management in Africa.

    • The title does not claim to offer a DSS, nor does the abstract or introduction. A key challenge for "socio-hydrogeology" is how to respond to situations where available information and problems are not as well structured as typically assumed for a DSS. 

Reviewer 2 Report

The author presents a model of groundwater management in which governance is put in the hands of farmers, water users, and stakeholders rather than being a top-down set of regulations provided by the government. The paper is well written and offers a useful overview of groundwater management that is useful for regulator as well as hydrogeologists. I suggest that the paper to be accepted; however, as a hydrogeologist coming from a technical rather than managerial angle of dealing with groundwater flows, I had the following observations that I would like to be addressed by the author before publication:

  • While the paper is providing a model, it is not very clear to whom this model is really addressed. Is it for farmers with limited resources, whether capital, technical or not very well organised, or is it for farmers who are benefiting from a well governed country?
  • The paper stresses on the first type of farmers where there is a lack of groundwater governance but at the same time it shows that these farmers rely on important amount of information that are provided to them by governmental bodies such as understanding the type of aquifers and their characteristics, research into climate change, remote sensing data, water quality data, etc. There is some contradiction here.
  • The paper touches on problems related to the application of the proposed model, one is resolving dispute between the farmers. However, it resorts to government power and tribunal to resolve the problem and that could be lacking in the first place.
  • The model does not discuss the problems associated with growth, mainly population growth, and pressure on resources. While the cooperation model between farmers works, where resources are available, the situation may become different if the land becomes crowded and renewable resources become inadequate. There is a need for management at a high level for a sustainable development.
  • The model is trying to replace the role of the government, as the regulations produced at the top are not followed. The paper indicates that the model will offer a solution as the stakeholders will learn from each other and cooperate by sharing water, rotating crops, etc. However, there are a couple of important issues that has not been discussed and that can be addressed only by the government. For example, ensuring a market that will take produced crops from all farmers and not the lucky ones only, compensating for failed crops, etc.

I suggest that the author addresses these point more specifically if possible but I think the paper can be published in its current form.

Author Response

The author presents a model of groundwater management in which governance is put in the hands of farmers, water users, and stakeholders rather than being a top-down set of regulations provided by the government. The paper is well written and offers a useful overview of groundwater management that is useful for regulator as well as hydrogeologists. I suggest that the paper to be accepted; however, as a hydrogeologist coming from a technical rather than managerial angle of dealing with groundwater flows, I had the following observations that I would like to be addressed by the author before publication:

  • Thank you for the helpful comments. As described in more detail below, the comments have been considered in revising the manuscript, particularly to clarify specific points. Specific changes in the manuscript can also be seen using Track Changes.
  • While the paper is providing a model, it is not very clear to whom this model is really addressed. Is it for farmers with limited resources, whether capital, technical or not very well organised, or is it for farmers who are benefiting from a well governed country?
    • Added sentence at end of first paragraph of Section 1.3:
      • This paper is particularly concerned with ways to improve groundwater governance under conditions with limited information, administrative capacity, resources, and political support for groundwater governance.
    • Second paragraph of section 3, second sentence also revised:
      • Analysis can offer some insights into actions that influence the feasibility of various governance actions, particularly for situations where there is limited potential for top-down regulation or for resource-intensive forms of co-management.
    • The paper stresses on the first type of farmers where there is a lack of groundwater governance but at the same time it shows that these farmers rely on important amount of information that are provided to them by governmental bodies such as understanding the type of aquifers and their characteristics, research into climate change, remote sensing data, water quality data, etc. There is some contradiction here.
      • Section 4 discusses ways to support better groundwater governance. The first paragraph states that this can come from governments, non-government organizations, farmer groups, and others. Information is discussed extensively in subsection 4.1.
    • The paper touches on problems related to the application of the proposed model, one is resolving dispute between the farmers. However, it resorts to government power and tribunal to resolve the problem and that could be lacking in the first place.
      • Much of the paper discusses things that farmers may be able to do among themselves, without requiring dispute resolution by government. However, government can certainly also play a useful role. Second paragraph of Section 4.6 revised.
        • … ways in which regulations may provide a clear framework for private action, to reduce uncertainties and provide ways of resolving disputes through courts, mediation by local authorities, and other means.
      • The model does not discuss the problems associated with growth, mainly population growth, and pressure on resources. While the cooperation model between farmers works, where resources are available, the situation may become different if the land becomes crowded and renewable resources become inadequate. There is a need for management at a high level for a sustainable development.
        • Section 1.2 second to last sentence revised to further clarify need for better institutions, which could come from government in some cases, or through self-organized activities by water users, which is the main topic of the paper.
          • Institutions to cope with impacts of intensified water use (externalities) on others such as neighbors and downstream water users are often absent or ineffective
        • The model is trying to replace the role of the government, as the regulations produced at the top are not followed. The paper indicates that the model will offer a solution as the stakeholders will learn from each other and cooperate by sharing water, rotating crops, etc. However, there are a couple of important issues that has not been discussed and that can be addressed only by the government. For example, ensuring a market that will take produced crops from all farmers and not the lucky ones only, compensating for failed crops, etc.
          • The concept is not to “replace” government, but to consider more optimal combinations of government, commercial, and community activities, especially where resources are limited. Conclusions revised:
            • The ideas presented here offer alternatives for a diverse mix of institutional solutions, on their own or in combination with co-management and government regulation.
          • Constraints and opportunities related to markets are mentioned in the first paragraph, third sentence of section 1.2, at the end of section 1.3, and at the end of section 4.4.
            • “Even within current conceptions of WUAs in water governance there are many options for targeting efforts on specific activities and results, including better-informed participatory planning, field-level water service delivery, and linkage with input and output markets (value chains).”
          • Crop insurance is beyond the scope of this paper but could be included in the measures mentioned in Section 1.2.

I suggest that the author addresses these point more specifically if possible but I think the paper can be published in its current form.

  • As explained above, the paper has been revised to respond to the points above, particularly to present ideas more clearly.

Reviewer 3 Report

General comments

This is a well-written and valuable contribution and I look forward to citing it in future research publications.

While I note that Water has no restrictions on the length of manuscripts, the author guidelines also require that the text be ‘concise and comprehensive’. Overall, I think the manuscript would benefit from revision (and perhaps structuring) to ensure that, throughout, the text is less discursive and that statements (which often read as opinions) are better supported by the literature. The middle section, while relevant and of interest, needs to be better integrated into the overall concept of the ‘ladder of participation’.

Specific comments

p.4 ‘Section 1.5 Organization of the Paper’ ... this subheading appears out of place. In any case, this subsection is also a little heavy going, given the flow of the previous discussion. Perhaps you can just add a sentence or two to the previous para that point to these topics but without placing emphasis on the structure of the paper.

p.6 Table 1 ... this might be better presented graphically; for example, see Fig.1 in Hewitt et al. 2017 https://www.nature.com/articles/nclimate3378

p.5-6 Section 2 ... discussion in paras 2-4 is also heavy going and not well referenced.

p.7 Section 3 ... check sentence construction here and provide examples: ‘These may well be oriented towards solving specific problems or realizing particular opportunities, “reactive” rather than the kind of pro-active, integrated, and com-prehensive governance recommended as the ideal in much recent literature on ground-water governance.’

p.7 Final para of Section 3 ... the statements/discussion here needs to be better constructed/supported by examples and referenced.

p. 16 ‘Atomistic anarchy’ is a term only introduced in the conclusion and unexplained ... why? Reference to Table 2 (and discussion of these points) should be in the body of the paper rather than the conclusion.

Author Response

General comments

This is a well-written and valuable contribution and I look forward to citing it in future research publications.

  • Thank you for your comments.
  • While I note that Waterhas no restrictions on the length of manuscripts, the author guidelines also require that the text be ‘concise and comprehensive’. Overall, I think the manuscript would benefit from revision (and perhaps structuring) to ensure that, throughout, the text is less discursive and that statements (which often read as opinions) are better supported by the literature. The middle section, while relevant and of interest, needs to be better integrated into the overall concept of the ‘ladder of participation’.
    • Sentence added to first paragraph of Section 3.
      • These examples illustrate the potential for farmer-led and community-based activities in the extended ladder of participation.
    • Further review of literature and addition of specific references would be interesting, but is beyond the resources available for preparation of this paper.
  • Specific comments
  • 4 ‘Section 1.5 Organization of the Paper’ ... this subheading appears out of place. In any case, this subsection is also a little heavy going, given the flow of the previous discussion. Perhaps you can just add a sentence or two to the previous para that point to these topics but without placing emphasis on the structure of the paper.
    • The section has been retitled “Structure and Synopsis of the Paper to clarify the purpose and content. It is intended to provide a brief overview of what’s ahead, and for some readers to help them decide if they want to continued reading, or to get some key takeway messages without going through the rest of the paper. It could be deleted if this conflicts with MDPI’s style.
  • 6 Table 1 ... this might be better presented graphically; for example, see Fig.1 in Hewitt et al. 2017 https://www.nature.com/articles/nclimate3378
    • Thanks for the reference. However, the source is paywalled. Due to COVID I can’t go to to the libraries where I might be able to get access. I’ve contacted the author to request the paper but did not receive a response.
    • The earlier paper reviewed a range of participation scales, as ladders, tables, spectrum, and other diagrams and concluded by using the ladder. This paper has added the three broad categories on the left (in all caps) to give a sense of the larger differences. Horizontal line above level 3 seems to have been lost in editing and has been restore to clarify the three groupings. Table content has also been left-justified to make it easier to read and make the structure clearer.
  • 5-6 Section 2 ... discussion in paras 2-4 is also heavy going and not well referenced.
    • Most of the more complex material has been deleted, given the audience for this article, in contrast to the conference where the paper was originally presented. Summary sentence added at end of first paragraph of section 2: "Levels of participation represent a way to provide a simple summary of a complex set of relationships, which could also be analyzed in more detail in terms of different kinds of social network linkages and associated forms of power and influence
  • 7 Section 3 ... check sentence construction here and provide examples: ‘These may well be oriented towards solving specific problems or realizing particular opportunities, “reactive” rather than the kind of pro-active, integrated, and com-prehensive governance recommended as the ideal in much recent literature on ground-water governance.’
    • Sentence split up. Wording about reactive vs pro-active deleted. It’s related to section 1.3 on limitations of current concepts of groundwater governance, but doesn’t seem crucial here.
  • 7 Final para of Section 3 ... the statements/discussion here needs to be better constructed/supported by examples and referenced.
    • Citations added to Shah and Blomquist
  • 16 ‘Atomistic anarchy’ is a term only introduced in the conclusion and unexplained ... why? Reference to Table 2 (and discussion of these points) should be in the body of the paper rather than the conclusion.
    • Changed anarchy to “individual action.” Atomistic was used in the introduction.
    • Thank you again for the helpful comments.

Round 2

Reviewer 1 Report

A BRIEF SUMMARY

The paper titled Polycentric Solutions for Groundwater Governance in Sub-Saharan Africa: Encouraging Institutional Artisanship in an Extended Ladder of Participationpresents a good topic for readers of this Journal. However, several lacks emerge after reading the paper. Below is the list of serious lacks to justify my decision to reject the manuscript.

  • Lack of novelty. This manuscript represents a Case Study with some photos and tables, without graphs and numerical processing.
  • Lack of an adequate method description.
  • Lack of results description. In a scientific paper, the results are represented with graphs and/or maps.
  • Too little bibliography for this type of work, on a so broad topic. I strongly suggest that the authors try to add some more references especially in the "part 1 (introduction)" of the paper to make the foundation for the arguments stronger.

SPECIFIC COMMENTS:

Abstract: It is too long. (Usually max 200 words).